| Clinical Microbiology | Research Article

# Gram staining decipherment using an artificial intelligence-powered smartphone-based application

Kei Yamamoto,[1] Goh Ohji,[2] Isao Miyatsuka,[3] Kei Furui-Ebisawa,[2] Ataru Moriya,[4] Hidetoshi Nomoto,[1] Masami Kurokawa,[4] Kenichiro Ohnuma,[5] Mari Kusuki,[5] Norio Ohmagari[1]

**ABSTRACT**  Gram staining provides rapid microbiological information that may assist in empirical antimicrobial selection; however, the results are often interpreted by microbiological specialists who are not always available. Therefore, we developed a computer-aided diagnosis system using artificial intelligence trained on microscopic images of Gram-stained urine, captured with an iPhone, using the Bartholomew and Mittwer method. The system interprets Gram-stained urine samples and classifies bacterial morphology (Class 1: 7 predefined morphology categories) and 17 predefined species-level categories (Class 2). In this retrospective observational study, five imaging devices and two staining methods (Bartholomew and Mittwer, Favor) were compared. Urine specimens were collected from two hospitals between 1 April and 31 December 2022. Validation images were generated using five devices (four smartphones and one microscopic camera). We used a micrometer with microscopy with all smartphones; some iPhone images were taken without a micrometer. Favor staining was only imaged using an iPhone without the micrometer. Image data sets were generated from 433 clinical and 17 spiked samples. The overall accuracy was 0.804 for Class 1 and 0.640 for Class 2. Images taken by the microscopic camera had the highest accuracy and kappa coefficient, whereas the AQUOS smartphone had the lowest accuracy and kappa coefficient. The accuracy of images created without a micrometer was 0.885 for Class 1 and 0.666 for Class 2. The Bartholomew and Mittwer method had better accuracy and a better kappa coefficient. Overall, accuracy depended on the staining method used in the training data, not on the imaging device.

**IMPORTANCE** Gram staining provides rapid information on both the site of infection and likely pathogens, guiding empirical antimicrobial selection. However, interpretation requires infectious disease expertise, which is not always available. We developed an artificial intelligence-based diagnostic support system trained on iPhone images of Gram-stained urine using the Bartholomew and Mittwer method to classify bacterial morphology (Class 1) and inferred species (Class 2). To provide essential baseline data on factors influencing accuracy and reliability, we compared Gram-stained urine images from two hospitals obtained with five imaging devices and two staining methods. Microscopic camera images showed the highest accuracy, whereas an AQUOS smartphone showed the lowest. Images without a micrometer performed better, and the Bartholomew and Mittwer method outperformed the Favor method. Accuracy increased when confidence levels were higher. Our findings suggest that using the same staining method as the training data and avoiding micrometer noise are critical, while device differences are less influential.

**KEYWORDS**  artificial intelligence, computer-aided diagnosis, gram staining, urinary tract infection

**Peer Reviewer** Ahmad Ahmad, International Centre for Genetic Engineering and Biotechnology, New Delhi, Delhi, India

Address correspondence to Kei Yamamoto, yamamoto.k@jihs.go.jp.

K.Y. received research grants from Fujirebio Inc., Mizuho Medy Co. Ltd., VisGene Co. Ltd., Canon Medical Systems Corp., Sanyo Chemical Industries Ltd., CarbGeM Inc., Sysmex Corp., and Toyobo Co. Ltd. outside the submitted work and had patent rights to this CAD, but no income was generated from it. I.M. is an employee of CarbGeM Inc.

See the funding table on p. 13.

With bacterial culture testing, it can take several days to determine the appropriate antimicrobial agent. In contrast, Gram staining provides valuable microbiological information that can be obtained on the same day; however, it requires proficiency in evaluation (1), which is one of the barriers to its implementation. Therefore, the use of a computer-aided diagnostic (CAD) system to interpret Gram staining findings is anticipated.

Previous attempts have been made to decipher Gram staining findings using artificial intelligence (AI) based on deep learning for blood cultures (2, 3) and bacterial vaginosis (4, 5); however, to the best of our knowledge, there are no reports of the use of urine specimens. Therefore, we developed a smartphone-based CAD system for urine specimens to support the interpretation of Gram-stained urine findings in patients with suspected urinary tract infection and assist empirical antimicrobial selection, while reducing the implementation cost and enabling the use of the system in any setting. Although some studies question the utility of Gram staining of urine (6, 7), other studies suggest that this method has diagnostic utility and leads to a reduction in the use of broad-spectrum antibiotics (8, 9). Because urinary tract infections are very common, if our system leads to a reduction in the use of broad-spectrum antibiotics, it will have a significant impact on efforts to combat drug resistance. Additionally, urine specimens show less variation in quality than sputum, making them easier to evaluate. Importantly, there are various imaging devices, such as smartphones, and many known methods of Gram staining; thus, the hue and sharpness of the staining can vary depending on the staining method or reagent used. Therefore, we compared the results of the CAD-guided predictions between different imaging devices and staining methods. Furthermore, to investigate the influence of imaging devices and staining methods on microbiology specialist (MS) decipherment, some images were decoded by an MS.

## MATERIALS AND METHODS

### Study design

This retrospective observational study was conducted to evaluate the performance of a CAD system using residual clinical specimens from two tertiary care hospitals (the National Centre for Global Health and Medicine [NCGM] and Kobe University Hospital [KUH]). Because residual clinical specimens were used retrospectively, the requirement of individual informed consent was waived by providing an opt-out opportunity.

### AI model architecture and training

The CAD system was designed as a smartphone application employing a cloud-based AI model (Text S1). However, for the purposes of this study, images were analyzed using the same cloud-based model via a PC environment to ensure standardized evaluation.

The CAD system was developed using Python with the PyTorch and PyTorch Lightning frameworks. A total of 13,901 images derived from 1,350 slides collected between April 2020 and December 2023 were used for model development.

The following three models were constructed: (i) a Class 1 multi-label classifier for bacterial morphology (yeast, gram-positive cocci [GPC], gram-positive rods [GPR], gram-negative rods [GNR], gram-negative cocci [GNC], none, and polymicrobial); (ii) a species-level classifier for GPC categories; and (iii) a species-level classifier for GNR categories. The models were based on a fine-tuned ConvNeXt architecture pretrained on ImageNet.

To prevent data leakage, data splitting was performed on a per-slide basis. Model training used the AdamW optimizer with data augmentation, including RandAugment; mixup augmentation was applied to the Class 1 model. Multi-label evidential loss was used for morphology classification, and evidential loss was used for the species-level models to enable uncertainty estimation.

A fourfold cross-validation framework was adopted, and final model weights were optimized using the Model Soups method. During inference, images were

center-cropped and resized to 1,024 × 1,024 pixels. Confidence and uncertainty estimates were derived using the evidential learning framework.

## Inclusion criteria and collected information

Gram-stained slides of urine specimens sampled between 1 April and 30 June 2022 at NCGM and 1 April and 31 December 2022 at KUH were used. Clinical information, including the sampling date, specimen type, Gram staining findings, and species identification results, was collected.

After routine examination, samples with Gram staining and confirmed species identification were included. Samples were excluded if (i) a single organism was observed by microscopy despite more than one organism being identified by the bacterial culture (e.g., GNR was observed by microscopy, but *Pseudomonas aeruginosa* and *Escherichia coli* were isolated); (ii) if the identified organisms were clearly inconsistent with the microscopic findings (e.g., one GPR was observed by microscopy but *Escherichia coli* was isolated); or (iii) if the bacterial species could not be identified. These samples were excluded to prevent Gram-stained bacteria from being inappropriately linked to a Class 2 organism that clearly differed from the microscopic findings and to prevent cases in which the Gram-stain findings could not be linked to a single species.

## Classification

Two classifications were defined: one was based on the bacterial morphology (Class 1), and the other was based on the cultured bacterial species (Class 2).

The Class 1 model performed multi-label morphology classification and detected the presence or absence of predefined categories within a single microscopic field: gram-positive cocci GPC, GPR, GNR, GNC, yeast, polymicrobial findings, and no bacteria. The Class 2 model classified 17 predefined species-level categories based on culture-confirmed reference data (Table 1). The reference standard for both classifications was determined according to these predefined categories, and both CAD and microbiology specialists generated predictions accordingly.

## Image acquisition, devices, and data set

A single Gram-stained microscopic field was captured as one image under standardized magnification (×1,000, oil immersion). At the NCGM, image files were generated

**TABLE 1** Classifications[a]

| Code | Class 1 | Class 2 |
|---|---|---|
| U-01 | Yeast | *Candida* spp. |
| U-02 | GPC | GPC cluster |
| U-03 | GPC | *Enterococcus faecalis* |
| U-04 | GPC | *Enterococcus faecium* |
| U-05 | GPC | *Streptococcus agalactiae* |
| U-06 | GPC | Other GPC |
| U-07 | GPR | *Corynebacterium* spp. |
| U-08 | GNR | *Enterobacter cloacae* |
| U-09 | GNR | *Escherichia coli* |
| U-10 | GNR | *Klebsiella oxytoca* |
| U-11 | GNR | *Klebsiella pneumoniae* |
| U-12 | GNR | Other GNR Enterobacteriaceae |
| U-13 | GNR | *Pseudomonas aeruginosa* |
| U-14 | GNR | Other GNR glucose non-fermenting bacteria |
| U-15 | GNC | GNC |
| Poly | Polymicrobial | Polymicrobial |
| None | None | None |

[a]GPC, gram-positive cocci; GPR, gram-positive rod; GNC, gram-negative cocci; and GNR, gram-negative rod.

**TABLE 2** Number of target specimens extracted and images taken

| Actual number of samples from each institute | Number of samples included | Number of images per sample |
|---|---|---|
| 0–4 | 3[a] | 12 |
| 5–9 | 5 | 7 |
| 10–14 | 10 | 4 |
| 15–19 | 15 | 3 |
| ≥20 | 20 | 2 |

[a]When there were less than three available specimens, to incorporate three specimens, the specimens were supplemented with spiked specimens.

using five imaging devices: iPhone (iPhone Xs), Galaxy (Galaxy S20), Xperia (Xperia 5), AQUOS (AQUOS sense5G SH-M17), and a microscopic camera (DP23, Olympus) for specimens stained using the Bartholomew and Mittwer (B&M) method (Data set 1). At KUH, specimens were stained using both the B&M and Favor methods, and images were generated using an iPhone 12 without a micrometer (Data set 2). For Data set 1, images from the same sample were captured across multiple imaging devices to evaluate inter-device variability, resulting in a larger total number of images compared to Data set 2. The device position was adjusted using a NexyZ universal smartphone adapter (Celestron, CA, USA) to maximize the visible microscopic field within the camera frame after optimal focus was achieved using the microscope. Thereafter, images were captured using 1.9× optical zoom, ensuring a uniform field of view and scale across devices. The optical microscopes used were a BX53 (Olympus) at the NCGM and a CX41LF (Olympus) at KUH. All smartphone devices had rear cameras with at least 12 megapixels. For Data set 1, image size was standardized during acquisition using a micrometer (the scale was added only to the microscopic camera images). All images represented a single microscopic field and were subsequently resized to a fixed resolution during model inference to ensure consistency across devices.

Table 2 lists the number of images captured according to the number of samples in Class 2. For classifications with no more than three samples, spiked samples were generated using the ATCC standard strains (Text S2). In addition, 10 images were captured per sample from 15 to 50 samples randomly selected among samples with undetected bacteria during microscopy (labeled as "None"), and samples with two or more bacteria with different morphology (e.g., not applicable to cases where GPC clusters and chains were mixed, but applicable where GPC and GNC were mixed) were labeled as "Polymicrobial." The deciphering is presented in Text S3.

A total of 1,388 and 1,869 eligible samples were identified at the NCGM and KUH, respectively (Table S1).

## Comparison to MS

From these samples, three were randomly extracted from each Class 2 category, and three image files per sample were included for deciphering by an MS. Images randomly extracted from Data set 1 were used for KUH (four physicians and six technicians), and images randomly extracted from Data set 2 were used for 10 MSs (five physicians and five technicians) at the NCGM.

The MS included physicians and clinical laboratory technicians who were infectious disease specialists certified by the Japanese Association of Infectious Diseases and technologists in microbiology certified by the College of Laboratory Medicine in Japan or by Medical Technologists in Clinical Microbiology.

## Outcomes

The performance (accuracy) of the CAD classified as Class 1 and Class 2 was compared for each device or staining method in each data set as the primary outcome. Secondary

outcomes included the balanced accuracy, recall, precision, and harmonic mean (F1) for Class 1 and Class 2 categories. Recall, precision, balanced accuracy, and the F1 score were calculated using the following equations (10):

$$\text{Accuracy} = \frac{\text{True positive} + \text{True negative}}{\text{All}},$$

$$\text{Recall} = \frac{\text{True positive}}{\text{True positive} + \text{False negative}},$$

$$\text{Precision} = \frac{\text{True positive}}{\text{True positive} + \text{False positive}},$$

$$F_1 \text{ score} = 2 \times \frac{\text{Precision} \times \text{Recall}}{\text{Precision} + \text{Recall}},$$

$$\text{Balanced accuracy} = \frac{1}{2} \times \left( \frac{\text{True positive}}{\text{True positive} + \text{False negative}} + \frac{\text{True negative}}{\text{True negative} + \text{False positive}} \right).$$

In addition, validation of the "confidence value" and comparison with the decipherment performance of MS were included as secondary outcomes.

## Statistical analysis

The accuracy of the primary outcome is expressed as the point estimate and 95% confidence interval (CI), and paired comparisons were made using Fisher's exact probability test for accuracy; $P$-values were corrected using the Benjamini–Hochberg method for multiple comparisons. The significant probability was 5%.

Accuracy was evaluated for Class 1 and Class 2 in three strata of confidence values: less than the 25th percentile of the distribution, greater than the 25th percentile and less than the 50th percentile, and greater than the 50th percentile. The accuracy of each stratum was tested using the Cochran–Armitage test. The receiver operating characteristic (ROC) curve was drawn with prediction agreement for Class 1 and Class 2 as the objective variables and the confidence value as the independent variable. The area under the curve (AUC) and 95% CI were calculated using the DeLong method. For comparison with MS, the point estimates and 95% CIs of the accuracy of each imaging device image and staining method were calculated, and the recall, precision, F1 value, and balanced accuracy for each category were compared. R version 4.3.0 (R Foundation for Statistical Computing, Vienna, Austria) was used for statistical analysis.

## RESULTS

Of the eligible samples, 281 and 152 clinical samples were obtained from NCGM and KUH, respectively. A total of 6 spiked samples (3 each of GNC and other GNR glucose non-fermenting bacteria) and 11 spiked samples (3 GNC, 3 other GNR glucose non-fermenting bacteria, 2 *Enterococcus faecium*, 1 other GPC, and 2 *Klebsiella oxytoca*) were obtained from NCGM and KUH, respectively. Data set 1 included 7,940 images, and Data set 2 included 2,364 images for analysis.

## Performance of the CAD system

The accuracy for Data set 1 was 0.804 (0.795–0.813) for Class 1 and 0.640 (0.629–0.650) for Class 2; the confusion matrix is shown in Fig. 1. The results of each device are presented in Table 3 and Table S2a. In the Class 1 pairwise comparisons, there were significant differences between the microscopic camera and the other devices and between AQUOS and the other devices (Table S3a), whereas in Class 2, there were significant differences between the microscopic camera and AQUOS ($P < 0.001$) and Xperia ($P = 0.032$) (Table S3b).

The accuracy of Data set 2 was 0.885 (0.872–0.898) for Class 1 and 0.666 (0.647–0.685) for Class 2, and the confusion matrix is shown in Fig. 2. The results for each staining method are presented in Table 4 and Table S2b. Fisher's exact probability test showed

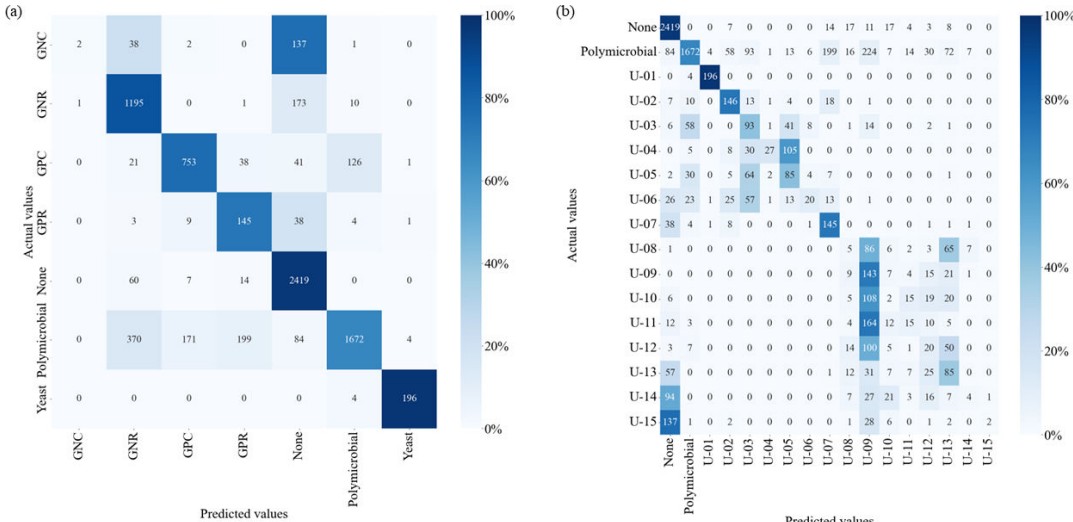

**FIG 1** Confusion matrix for Data set 1 predictions, classified by bacterial morphology. (a) Class 1. (b) Class 2. GPC, gram-positive cocci; GPR, gram-positive rod; GNC, gram-negative cocci; and GNR, gram-negative rod.

significant differences between Class 1 and Class 2, respectively ($P < 0.001$ and $P = 0.02$) (Table S3c and d).

## Confidence values

The distribution of confidence values is shown in Fig. S1. The medians for Class 1 were 0.835 and 0.854, and the medians for Class 2 were 0.740 and 0.732 for Data sets 1 and 2, respectively. As shown in Table 5, the accuracy tended to increase as the confidence value increased ($P < 0.001$). The trend of the F1 value was generally similar to that of Class 1, except for GNC and polymicrobial classifications. Conversely, F1 tended to decrease in steps for the polymicrobes in Data set 1 (Table 5). The ROC curves are shown in Fig. S2. The AUCs for Data set 1 for Class 1 and Class 2 were 0.680 (0.667–0.693) and 0.811 (0.801–0.821), respectively, and for Data set 2, they were 0.856 (0.835–0.878) and 0.887 (0.874–0.900), respectively.

## Comparison with MS

In Data set 1, both the MS and CAD were validated for each imaging device, with 510 predictions by the MS and 51 predictions by the CAD. The accuracy, recall, precision, F1 value, and balanced accuracy (Table 6) were evaluated for each device, and the CAD showed overall lower accuracy and kappa coefficients compared to the MS. The imaging device with the highest accuracy for CAD was a microscopic camera, which was the same for the MS.

In Data set 2, the MS was validated with 510 predictions for the Favor method and 1,530 predictions for the B&M method, while the CAD was validated with 51 and 153 predictions, respectively. In the evaluation of classification accuracy (Table 6), the MS system achieved higher accuracy and kappa coefficients with the Favor method, whereas the CAD system outperformed with the B&M method.

## DISCUSSION

In this study, we explored practical factors affecting AI-assisted interpretation of Gram-stained urine specimens in routine laboratory settings. Instead of focusing exclusively on algorithm development, we evaluated how imaging devices, staining methods, and image acquisition protocols affect classification performance and real-world implementation feasibility.

**TABLE 3** Classification performance for Class 1 by device (Data set 1)[a]

|  | All | DP23 | Galaxy | iPhone | AQUOS | Xperia |
|---|---|---|---|---|---|---|
| Accuracy | 0.80 | 0.84 | 0.81 | 0.81 | 0.76 | 0.80 |
| 95% CI | 0.80–0.81 | 0.82–0.86 | 0.79–0.83 | 0.79–0.83 | 0.74–0.79 | 0.78–0.82 |
| Kappa | 0.74 | 0.79 | 0.75 | 0.75 | 0.69 | 0.74 |
| Macro recall | 0.71 | 0.75 | 0.71 | 0.69 | 0.68 | 0.73 |
| Balanced accuracy |  |  |  |  |  |  |
| GNC | 0.51 | 0.50 | 0.50 | 0.50 | 0.51 | 0.51 |
| GNR | 0.90 | 0.91 | 0.91 | 0.92 | 0.86 | 0.89 |
| GPC | 0.87 | 0.93 | 0.83 | 0.82 | 0.87 | 0.91 |
| GPR | 0.85 | 0.89 | 0.86 | 0.82 | 0.79 | 0.87 |
| Multi | 0.82 | 0.86 | 0.84 | 0.84 | 0.77 | 0.80 |
| None | 0.94 | 0.95 | 0.94 | 0.94 | 0.92 | 0.95 |
| Yeast | 0.99 | 1.00 | 1.00 | 0.96 | 0.99 | 1.00 |
| F1 |  |  |  |  |  |  |
| GNC | 0.02 | NA | NA | NA | 0.05 | 0.05 |
| GNR | 0.78 | 0.83 | 0.80 | 0.79 | 0.73 | 0.75 |
| GPC | 0.78 | 0.83 | 0.73 | 0.75 | 0.77 | 0.84 |
| GPR | 0.49 | 0.50 | 0.48 | 0.50 | 0.44 | 0.50 |
| Multi | 0.77 | 0.83 | 0.80 | 0.79 | 0.70 | 0.75 |
| None | 0.90 | 0.92 | 0.90 | 0.91 | 0.86 | 0.90 |
| Yeast | 0.98 | 0.99 | 1.00 | 0.94 | 0.98 | 0.98 |
| Precision |  |  |  |  |  |  |
| GNC | 0.67 | NA | NA | NA | 0.50 | 1.00 |
| GNR | 0.71 | 0.81 | 0.74 | 0.69 | 0.66 | 0.66 |
| GPC | 0.80 | 0.77 | 0.79 | 0.86 | 0.77 | 0.83 |
| GPR | 0.37 | 0.36 | 0.35 | 0.40 | 0.35 | 0.37 |
| Multi | 0.92 | 0.95 | 0.90 | 0.87 | 0.94 | 0.97 |
| None | 0.84 | 0.88 | 0.85 | 0.87 | 0.76 | 0.84 |
| Yeast | 0.97 | 0.98 | 1.00 | 0.95 | 0.98 | 0.95 |
| Recall |  |  |  |  |  |  |
| GNC | 0.01 | 0.00 | 0.00 | 0.00 | 0.03 | 0.03 |
| GNR | 0.87 | 0.86 | 0.88 | 0.92 | 0.81 | 0.87 |
| GPC | 0.77 | 0.90 | 0.68 | 0.66 | 0.77 | 0.84 |
| GPR | 0.73 | 0.83 | 0.75 | 0.68 | 0.60 | 0.78 |
| Multi | 0.67 | 0.73 | 0.72 | 0.73 | 0.56 | 0.61 |
| None | 0.97 | 0.96 | 0.96 | 0.95 | 0.99 | 0.98 |
| Yeast | 0.98 | 1.00 | 1.00 | 0.93 | 0.98 | 1.00 |

[a]CI, confidence interval; GPC, gram-positive cocci; GPR, gram-positive rod; GNC, gram-negative cocci; and GNR, gram-negative rod; and NA, not applicable.

Although some devices showed notable differences—such as AQUOS, which demonstrated significantly lower accuracy, and the microscope camera DP23, which achieved higher accuracy and macro-recall with significant improvement in Class 1 classification—the overall impact of imaging devices on accuracy was limited. Notably, although images taken with an iPhone using the B&M staining method were included in the training data, the accuracy of CAD prediction did not necessarily depend on the imaging device used for training. In contrast, specimens stained with the Favor method yielded inferior results compared with those stained with the B&M method.

Although the factors that affect the classification ability of image AI are not fully understood, images other than the target object can affect classification (5). Narla et al. (11) found that the scale, aside from the target findings, was biased toward malignant tumors, resulting in higher discrimination performance. However, the use of micrometers in this study resulted in lower classification performance. Although it was unclear whether the addition of the micrometer affected the quality of the images, such as focus,

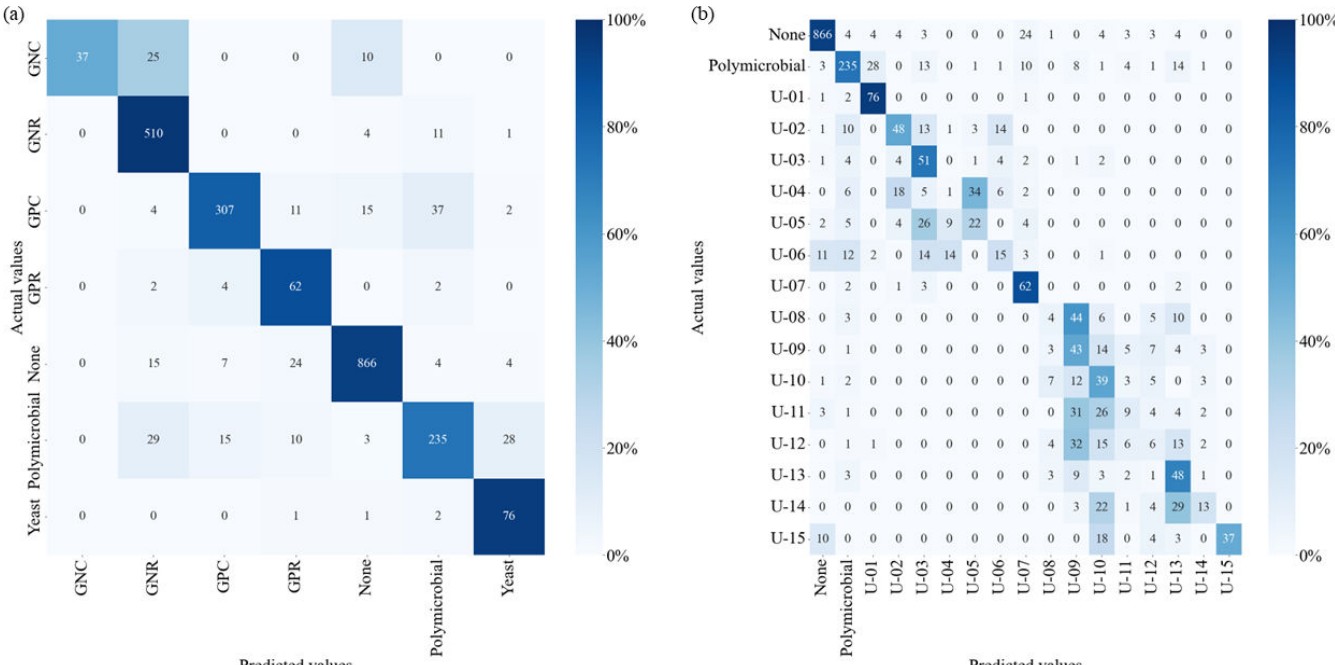

**FIG 2** Confusion matrix for Data set 2 predictions, classified by detailed bacterial species or groups. (a) Class 1. (b) Class 2. GPC, gram-positive cocci; GPR, gram-positive rod; GNC, gram-negative cocci; and GNR, gram-negative rod.

or whether the noise of the scale affected discrimination, the classification performance was clearly inferior to that of Data set 1 with the micrometer and Data set 2 without the micrometer (accuracy of Class 1 in images stained with B&M and captured with an iPhone: 0.81 [0.79–0.83] in Data set 1 vs 0.91 [0.90–0.93] in Data set 2). Unlike the micrometer, the microscopic camera has a scale that appears at a fixed location, making it easy to crop the image and remove the scale from the image information. This does not affect the focus of the Gram-stained image, which may have contributed to the better classification results in this study. In the future, when applying this system in practice, it may be necessary to standardize imaging methods to avoid including extraneous information—such as scales or other elements not directly related to the Gram-stained field—in the microscopic images.

The error rate of interpreting Gram staining findings (Class 1) in this study was approximately 10% at best, which was higher than that reported in a previous study (12). However, the distribution of bacterial species in our cohort differed from previous reports (12). As in that study, polymicrobial specimens and those without detectable bacteria were not included in the analysis. In our previous non-inferiority study of MS, the accuracy was approximately 80%; MS included both microbiologists and infectious disease physicians, but none showed individual results exceeding 95% (13). For microbiologists in Japan, a decision test for accuracy control was conducted, where the error rate was not as high (<5%) (data not shown). We believe that the reason for this was the determination of whether the image of a single field captured with a smartphone was polymicrobial. Interestingly, the prediction of polymicrobial samples by MS tended to be less accurate than other classifications. In the preparatory test, there was no difference in performance between the use of a microscope, three digital images of a microscopic finding, and one image of those used in the test to verify the MS discrimination performance; however, this test did not include polymicrobes (14). This may be because, in this study, MSs made their judgments based on single captured images rather than by directly manipulating the microscope. In the future, it will be necessary to verify the performance of MSs when they evaluate polymicrobial samples through direct microscopic examination.

**TABLE 4** Classification performance for Class 1 by staining method (Data set 2)[a]

|  | All | B&M | Favor |
|---|---|---|---|
| Accuracy | 0.89 | 0.91 | 0.86 |
| 95% CI | 0.87–0.90 | 0.90–0.93 | 0.84–0.88 |
| Kappa | 0.85 | 0.88 | 0.81 |
| Macro recall | 0.83 | 0.90 | 0.76 |
| Balanced accuracy |  |  |  |
| GNC | 0.76 | 1.00 | 0.51 |
| GNR | 0.96 | 0.98 | 0.95 |
| GPC | 0.90 | 0.92 | 0.88 |
| GPR | 0.93 | 0.91 | 0.96 |
| Multi | 0.85 | 0.83 | 0.87 |
| None | 0.96 | 0.98 | 0.94 |
| Yeast | 0.97 | 0.97 | 0.97 |
| F1 score |  |  |  |
| GNC | 0.68 | 1.00 | 0.05 |
| GNR | 0.92 | 0.95 | 0.89 |
| GPC | 0.87 | 0.88 | 0.85 |
| GPR | 0.70 | 0.74 | 0.66 |
| Multi | 0.77 | 0.76 | 0.78 |
| None | 0.95 | 0.97 | 0.93 |
| Yeast | 0.80 | 0.80 | 0.79 |
| Precision |  |  |  |
| GNC | 1.00 | 1.00 | 1.00 |
| GNR | 0.87 | 0.91 | 0.83 |
| GPC | 0.92 | 0.90 | 0.94 |
| GPR | 0.57 | 0.67 | 0.51 |
| Multi | 0.81 | 0.85 | 0.78 |
| None | 0.96 | 0.98 | 0.95 |
| Yeast | 0.68 | 0.69 | 0.68 |
| Recall |  |  |  |
| GNC | 0.51 | 1.00 | 0.03 |
| GNR | 0.97 | 0.98 | 0.96 |
| GPC | 0.82 | 0.86 | 0.78 |
| GPR | 0.89 | 0.83 | 0.94 |
| Multi | 0.73 | 0.69 | 0.78 |
| None | 0.94 | 0.97 | 0.91 |
| Yeast | 0.95 | 0.95 | 0.95 |

[a]CI, confidence interval; GPC, gram-positive cocci; GPR, gram-positive rod; GNC, gram-negative cocci; GNR, gram-negative rod; and B&M, Bartholomew and Mittwer.

Although this study produced classification results for a single image, it is possible that, in clinical situations, considering the judgment results of several images may be more accurate. In addition, the confidence value was calculated from the uncertainty, and the error rate was well below 5% for those with high confidence values (above the median distribution). The error rate may be reduced if physicians make a final decision based on the judgment of confidence levels.

The limitations of this study include the lack of validation for samples prepared by medical professionals other than MS or those at other institutions. Improving the accuracy of Gram staining by non-MS medical professionals remains a challenge in clinical practice, and the findings of this study may be difficult to generalize to samples prepared by non-MS medical professionals or to institutions without MS personnel. Factors such as staining performed by non-specialists and the handling of equipment warrant further investigation. In addition, variability in the microscopes used, their cleanliness, and differences in staining methods or pretreatment procedures across

**TABLE 5** Performance of classification stratified by the confidence value[a]

|  | Data set 1 | | | Data set 2 | | |
|---|---|---|---|---|---|---|
| **Class 1** | | | | | | |
| Confidence value | ≤0.76 | >0.76, ≤0.84 | >0.84 | ≤0.81 | >0.81, ≤0.85 | >0.85 |
| N | 1,986 | 2,004 | 3,950 | 592 | 596 | 1,176 |
| Accuracy | 0.69 | 0.73 | 0.90 | 0.66 | 0.92 | 0.98 |
| 95% CI | 0.67–0.71 | 0.71–0.75 | 0.89–0.91 | 0.62–0.70 | 0.90–0.94 | 0.97–0.99 |
| Kappa | 0.36 | 0.65 | 0.83 | 0.56 | 0.90 | 0.96 |
| F1 score | | | | | | |
| Polymicrobial | 0.83 | 0.70 | 0.57 | 0.77 | 0.84 | NA |
| None | 0.21 | 0.60 | 0.94 | 0.49 | 0.91 | 0.99 |
| GNC | 0.17 | NA | NA | 0.86 | NA | NA |
| GNR | 0.32 | 0.76 | 0.88 | 0.57 | 0.91 | 0.99 |
| GPC | 0.48 | 0.83 | 0.94 | 0.64 | 0.98 | 0.94 |
| GPR | 0.27 | 0.57 | 0.81 | 0.48 | 0.95 | 1.00 |
| Yeast | 0.57 | 0.97 | 1.00 | 0.36 | 0.97 | 0.95 |
| **Class 2** | | | | | | |
| Confidence value | ≤0.54 | >0.54, ≤0.74 | >0.74 | ≤0.54 | >0.54, ≤0.73 | >0.73 |
| N | 1,997 | 1,981 | 3,962 | 591 | 289 | 1,182 |
| Accuracy | 0.29 | 0.57 | 0.85 | 0.26 | 0.50 | 0.95 |
| 95% CI | 0.27–0.31 | 0.55–0.59 | 0.84–0.87 | 0.23–0.30 | 0.46–0.55 | 0.94–0.96 |
| Kappa | 0.21 | 0.42 | 0.72 | 0.20 | 0.46 | 0.88 |
| F1 score | | | | | | |
| Polymicrobial | 0.54 | 0.80 | 0.88 | 0.50 | 0.72 | 0.91 |
| None | NA | 0.20 | 0.92 | 0.06 | 0.41 | 0.98 |
| U-01 | 0.50 | 0.50 | 0.99 | NA | 0.31 | 0.89 |
| U-02 | 0.46 | 0.80 | NA | 0.52 | 0.67 | NA |
| U-03 | 0.32 | 0.35 | NA | 0.35 | 0.61 | NA |
| U-04 | 0.38 | NA | NA | 0.04 | NA | NA |
| U-05 | 0.26 | 0.45 | NA | 0.26 | 0.40 | NA |
| U-06 | 0.29 | 0.03 | NA | 0.22 | 0.31 | NA |
| U-07 | 0.10 | 0.28 | 0.62 | NA | 0.39 | 0.88 |
| U-08 | 0.04 | 0.03 | NA | 0.09 | NA | NA |
| U-09 | 0.25 | 0.25 | NA | 0.34 | 0.23 | NA |
| U-10 | NA | 0.06 | NA | 0.06 | 0.56 | NA |
| U-11 | 0.09 | 0.16 | NA | 0.18 | 0.13 | NA |
| U-12 | 0.16 | NA | NA | 0.10 | 0.09 | NA |
| U-13 | 0.16 | 0.46 | NA | 0.37 | 0.54 | NA |
| U-14 | 0.08 | 0.07 | NA | 0.24 | 0.29 | NA |
| U-15 | NA | 0.31 | NA | NA | 0.88 | 0.75 |

[a]CI, confidence interval; GPC, gram-positive cocci; GPR, gram-positive rod; GNC, gram-negative cocci; GNR, gram-negative rod; and NA, not applicable.

facilities may also influence performance. The former factors may act as sources of noise, while the latter may contribute to reduced classification accuracy, as observed in this study. Therefore, validation under diverse conditions and the accumulation of additional data are required. Finally, to enable linkage based on culture testing, several specimens that may arise in actual clinical practice were excluded. Consequently, future studies are needed to examine how the exclusions impact real-world applications.

**TABLE 6** Class 1 classification performance for the microbiology specialists and computer-aided diagnostic system[a]

| | | | GNC | GNR | GPC | GPR | Poly | None | Yeast |
|---|---|---|---|---|---|---|---|---|---|
| **Microbiology specialists** | | | | | | | | | |
| Data set 1 | | | | | | | | | |
| iPhone | Accuracy | Precision | 0.4 | 0.87 | 0.89 | 0.5 | 0.64 | 0.51 | 0.91 |
| | 0.79 (0.75–0.83) | Recall | 0.33 | 0.97 | 0.72 | 0.20 | 0.60 | 0.97 | 1.00 |
| | Kappa | F1 | 0.36 | 0.91 | 0.80 | 0.29 | 0.62 | 0.67 | 0.95 |
| | 0.71 | BA | 0.65 | 0.93 | 0.84 | 0.59 | 0.79 | 0.95 | 1.00 |
| AQUOS | Accuracy | Precision | 0.56 | 0.92 | 0.87 | 0.30 | 0.59 | 0.45 | 0.97 |
| | 0.79 (0.75–0.82) | Recall | 0.30 | 0.93 | 0.78 | 0.27 | 0.57 | 0.87 | 1.00 |
| | Kappa | F1 | 0.39 | 0.92 | 0.82 | 0.28 | 0.58 | 0.59 | 0.98 |
| | 0.71 | BA | 0.64 | 0.94 | 0.87 | 0.61 | 0.77 | 0.90 | 1.00 |
| Galaxy | Accuracy | Precision | 0.35 | 0.87 | 0.88 | 0.38 | 0.70 | 0.49 | 1.00 |
| | 0.78 (0.75–0.82) | Recall | 0.27 | 0.93 | 0.73 | 0.20 | 0.77 | 0.97 | 0.97 |
| | Kappa | F1 | 0.30 | 0.90 | 0.80 | 0.26 | 0.73 | 0.65 | 0.98 |
| | 0.70 | BA | 0.62 | 0.91 | 0.85 | 0.59 | 0.87 | 0.95 | 0.98 |
| Xperia | Accuracy | Precision | 0.33 | 0.84 | 0.87 | 0.31 | 0.54 | 0.48 | 0.94 |
| | 0.75 (0.71–0.79) | Recall | 0.30 | 0.89 | 0.74 | 0.13 | 0.50 | 1.00 | 0.97 |
| | Kappa | F1 | 0.32 | 0.86 | 0.80 | 0.19 | 0.52 | 0.65 | 0.95 |
| | 0.66 | BA | 0.63 | 0.88 | 0.85 | 0.56 | 0.74 | 0.97 | 0.98 |
| DP23 | Accuracy | Precision | 0.67 | 0.92 | 0.92 | 0.59 | 0.59 | 0.67 | 1.00 |
| | 0.85 (0.82–0.88) | Recall | 0.53 | 0.97 | 0.81 | 0.53 | 0.67 | 0.97 | 1.00 |
| | Kappa | F1 | 0.59 | 0.94 | 0.87 | 0.56 | 0.63 | 0.79 | 1.00 |
| | 0.80 | BA | 0.76 | 0.96 | 0.89 | 0.76 | 0.82 | 0.97 | 1.00 |
| Data set 2 | | | | | | | | | |
| B&M | Accuracy | Precision | 0.77 | 0.91 | 0.87 | 0.95 | 0.67 | 0.63 | 0.84 |
| | 0.85 (0.84–0.87) | Recall | 0.48 | 0.97 | 0.92 | 0.64 | 0.33 | 0.97 | 0.74 |
| | Kappa | F1 | 0.59 | 0.94 | 0.89 | 0.77 | 0.44 | 0.76 | 0.79 |
| | 0.80 | BA | 0.73 | 0.95 | 0.93 | 0.82 | 0.66 | 0.97 | 0.87 |
| Favor | Accuracy | Precision | 0.52 | 0.89 | 0.91 | 0.89 | 0.59 | 0.37 | 0.78 |
| | 0.80 (0.76–0.83) | Recall | 0.47 | 0.95 | 0.74 | 0.83 | 0.53 | 0.67 | 0.70 |
| | Kappa | F1 | 0.49 | 0.92 | 0.82 | 0.86 | 0.56 | 0.48 | 0.74 |
| | 0.72 | BA | 0.72 | 0.93 | 0.85 | 0.91 | 0.76 | 0.80 | 0.84 |
| **Computer-aided diagnostic system** | | | | | | | | | |
| Data set 1 | | | | | | | | | |
| iPhone | Accuracy | Precision | NA | 0.83 | 0.91 | 0.00 | 0.33 | 0.60 | 1.00 |
| | 0.75 (0.60–0.86) | Recall | 0.00 | 0.95 | 0.67 | 0.00 | 0.67 | 1.00 | 1.00 |
| | Kappa | F1 | NA | 0.89 | 0.77 | NA | 0.44 | 0.75 | 1.00 |
| | 0.65 | BA | 0.50 | 0.91 | 0.82 | 0.48 | 0.79 | 0.98 | 1.00 |
| AQUOS | Accuracy | Precision | NA | 1.00 | 1.00 | 0.33 | 0.40 | 0.21 | 1.00 |
| | 0.69 (0.54–0.81) | Recall | 0.00 | 0.86 | 0.53 | 0.33 | 0.67 | 1.00 | 1.00 |
| | Kappa | F1 | NA | 0.92 | 0.70 | 0.33 | 0.50 | 0.35 | 1.00 |
| | 0.60 | BA | 0.50 | 0.93 | 0.77 | 0.65 | 0.80 | 0.89 | 1.00 |
| Galaxy | Accuracy | Precision | NA | 1.00 | 0.83 | 0.50 | 0.50 | 0.20 | 1.00 |
| | 0.75 (0.60–0.86) | Recall | 0.00 | 0.95 | 0.67 | 0.33 | 0.67 | 0.67 | 1.00 |
| | Kappa | F1 | NA | 0.98 | 0.74 | 0.40 | 0.57 | 0.31 | 1.00 |
| | 0.66 | BA | 0.50 | 0.98 | 0.81 | 0.66 | 0.81 | 0.75 | 1.00 |
| Xperia | Accuracy | Precision | NA | 0.95 | 0.91 | 0.25 | 0.50 | 0.33 | 1.00 |
| | 0.75 (0.60–0.86) | Recall | 0.00 | 0.90 | 0.67 | 0.33 | 0.67 | 1.00 | 1.00 |
| | Kappa | F1 | NA | 0.93 | 0.77 | 0.29 | 0.57 | 0.50 | 1.00 |
| | 0.66 | BA | 0.50 | 0.94 | 0.82 | 0.64 | 0.81 | 0.94 | 1.00 |
| DP23 | Accuracy | Precision | NA | 0.95 | 0.93 | 0.67 | 0.40 | 0.50 | 1.00 |
| | 0.82 (0.69–0.92) | Recall | 0.00 | 0.90 | 0.87 | 0.67 | 0.67 | 1.00 | 1.00 |

*(Continued on next page)*

**TABLE 6** Class 1 classification performance for the microbiology specialists and computer-aided diagnostic system[a] (*Continued*)

| | | | GNC | GNR | GPC | GPR | Poly | None | Yeast |
|---|---|---|---|---|---|---|---|---|---|
| | Kappa | F1 | NA | 0.93 | 0.90 | 0.67 | 0.50 | 0.67 | 1.00 |
| | 0.76 | BA | 0.50 | 0.94 | 0.92 | 0.82 | 0.80 | 0.97 | 1.00 |
| Data set 2 | | | | | | | | | |
| B&M | Accuracy | Precision | 1.00 | 0.97 | 0.91 | 0.89 | 0.38 | 0.90 | 1.00 |
| | 0.92 (0.86–0.95) | Recall | 1.00 | 0.97 | 0.93 | 0.89 | 0.33 | 1.00 | 0.89 |
| | Kappa | F1 | 1.00 | 0.97 | 0.92 | 0.89 | 0.35 | 0.95 | 0.94 |
| | 0.88 | BA | 1.00 | 0.97 | 0.95 | 0.94 | 0.65 | 1.00 | 0.94 |
| Favor | Accuracy | Precision | NA | 0.83 | 1.00 | 0.50 | 0.29 | 0.40 | 1.00 |
| | 0.73 (0.58–0.84) | Recall | 0.00 | 0.90 | 0.60 | 0.67 | 0.67 | 0.67 | 1.00 |
| | Kappa | F1 | NA | 0.86 | 0.75 | 0.57 | 0.40 | 0.50 | 1.00 |
| | 0.63 | BA | 0.50 | 0.89 | 0.80 | 0.81 | 0.78 | 0.80 | 1.00 |

[a]GPC, gram-positive cocci; GPR, gram-positive rod; GNC, gram-negative cocci; GNR, gram-negative rod; BA, balanced accuracy; B&M, Bartholomew and Mittwer; and NA, not applicable.

## Conclusion

In CAD, where non-inferiority to MS has been proved, we verified performance differences based on imaging devices and staining methods. Except for one device, no significant differences in classification accuracy were observed between imaging devices, and there was no dependence on the device used to generate the training data. In contrast, staining methods showed a tendency to influence performance depending on the method used to create the training data. In future examinations, extraneous noise in the microscopic field of view (e.g., micrometers) should be minimized, and results with low confidence values should be interpreted with caution.

## ACKNOWLEDGMENTS

This research was supported by AMED under grant number JP23hk0102076 (to K.Y.) and JP24he2932005 (to K.Y.). The funders had no role in the study design, data collection and interpretation, or the decision to submit the work for publication.

The authors express their deepest gratitude to Masami Yoshino for managing the imaging data used in this study. We also thank Kayoko Hayakawa, Taketomo Maruki, Yutaro Akiyama, Yuki Moriyama, Ayano Motohashi, Kazuhisa Mezaki, Saeko Kinpara, Taiji Koyama, Kenta Iijima, Saori Kobayashi, Sayaka Matsumoto, Masako Nishida, and Nami Ishida for their help in deciphering the Gram staining. We also thank the staff at NCGM and KUH, including Yoshihiko Hirayama, Mayumi Kawakami, Motoko Ishida, Noriko Iwamoto, Gen Yamada, Satoshi Ide, Ayako Okuhama, Akiho Sugita, Nozomu Tsurumaki, Sei Ueda, Wataru Ochi, Reo Iguma, Kaito Mimura, Ryo Sakuma, Mayu Hiraiwa, Shunsuke Matsuoka, Akiko Hamana, Noriko Tomita, and Moto Kimura, for their cooperation.

K.Y., G.O., K.F.-E., and N.O. conceptualized the study; K.Y., A.M., M. Kurokawa, G.O., K.F.-E., K.O., M. Kusuki, and I.M. curated the data; K.Y. performed formal analysis; K.Y., G.O., and I.M. acquired funding; K.Y., G.O., K.F.-E., and I.M. conducted the investigation; K.Y., A.M., G.O., K.F.-E., and I.M. designed the methodology; K.Y., G.O., I.M., and N.O. contributed to project administration; A.M., M. Kurokawa, K.O., M. Kusuki, and I.M. provided resources; I.M. helped with software; N.O. supervised the study; K.Y. visualized the study and wrote the original draft. G.O., I.M., K.F.-E., A.M., H.N., M. Kurokawa, K.O., M. Kusuki, and N.O. reviewed and edited the manuscript.

## AUTHOR AFFILIATIONS

[1]Disease Control and Prevention Centre, National Centre for Global Health and Medicine, Japan Institute for Health Security, Tokyo, Japan
[2]Department of Infectious Disease, Kobe University Hospital, Kobe, Japan
[3]CarbGeM Inc., Tokyo, Japan

$^4$Department of Clinical Laboratory, National Centre for Global Health and Medicine, Japan Institute for Health Security, Tokyo, Japan
$^5$Department of Clinical Laboratory, Kobe University Hospital, Kobe, Japan

## PRESENT ADDRESS

Ataru Moriya, Department of Clinical Laboratory, Ibaraki Higashi National Hospital, Ibaraki, Japan

## AUTHOR ORCIDs

Kei Yamamoto  http://orcid.org/0000-0001-6218-523X
Norio Ohmagari  https://orcid.org/0000-0002-4622-8970

## FUNDING

| Funder | Grant(s) | Author(s) |
| --- | --- | --- |
| Japan Agency for Medical Research and Development | JP23hk0102076, JP24he2932005 | Kei Yamamoto |

## AUTHOR CONTRIBUTIONS

Kei Yamamoto, Conceptualization, Data curation, Formal analysis, Funding acquisition, Investigation, Methodology, Project administration, Visualization, Writing – original draft | Goh Ohji, Conceptualization, Data curation, Funding acquisition, Investigation, Methodology, Project administration, Writing – review and editing | Isao Miyatsuka, Data curation, Funding acquisition, Investigation, Methodology, Project administration, Resources, Software, Writing – review and editing | Kei Furui-Ebisawa, Conceptualization, Data curation, Investigation, Methodology, Writing – review and editing | Ataru Moriya, Data curation, Methodology, Resources, Writing – review and editing | Hidetoshi Nomoto, Writing – review and editing | Masami Kurokawa, Data curation, Resources, Writing – review and editing | Kenichiro Ohnuma, Data curation, Resources, Writing – review and editing | Mari Kusuki, Data curation, Resources, Writing – review and editing | Norio Ohmagari, Conceptualization, Project administration, Supervision, Writing – review and editing

## DATA AVAILABILITY

All data associated with this study are presented in the paper or Supplemental material and are available from the authors upon reasonable request. Individual-level, de-identified participant data are not available for publication.

## ETHICS APPROVAL

The study was approved by the Institutional Review Board of the NCGM (NCGM-S-004480-02).

## ADDITIONAL FILES

The following material is available online.

### Supplemental Material

**Supplemental material (Spectrum03076-25-s0001.docx).** Text S1 to S3, Tables S1 to S3, and Fig. S1 and S2.

## Open Peer Review

**PEER REVIEW HISTORY (review-history.pdf).** An accounting of the reviewer comments and feedback.

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
