## [Reviewer comments · Microbiology Spectrum]

Microbiology Spectrum

Gram Staining Decipherment using an Artificial Intelligence-Powered Smartphone-Based Application

Kei Yamamoto, Goh Ohji, Isao Miyatsuka, Kei Ebisawa, Ataru Moriya, Hidetoshi Nomoto, Masami Kurokawa, Kenichiro Ohnuma, Mari Kusuki, and Norio Ohmagari

Corresponding Author(s): Kei Yamamoto, Kokuritsu Kokusai Iryo Kenkyu Center Byoin

Review Timeline:

Submission Date:	September 24, 2025
Editorial Decision:	November 10, 2025
Revision Received:	November 24, 2025
Editorial Decision:	February 21, 2026
Revision Received:	February 26, 2026
Accepted:	March 28, 2026

Editor: Aditya Padhi

Reviewer(s): Disclosure of reviewer identity is with reference to reviewer comments included in decision letter(s). The following individuals involved in review of your submission have agreed to reveal their identity: AHMAD AHMAD (Reviewer #5)

Transaction Report:

DOI: <https://doi.org/10.1128/spectrum.03076-25>

Re: Spectrum03076-25 (Gram Staining Decipherment using an Artificial Intelligence-Powered Smartphone-Based Application)

Dear Dr. Kei Yamamoto:

Thank you for the privilege of reviewing your work. Below you will find my comments, instructions from the Spectrum editorial office, and the reviewer comments.

Revision Guidelines

Sincerely,
Aditya K. Padhi
Editor
Microbiology Spectrum

Reviewer #2 (Comments for the Author):

The authors present a comprehensive study examining the performance of gram stains using a variety of artificial intelligence based methods. The study is well structured and written. A few questions:

1; What was the reasoning behind targeting gram stains from urine samples? Multiple studies have shown that routine gram stains from urine samples have limited clinical utility and in many settings, labs no longer routinely perform gram stains from

urine samples.

2; It is unclear from the document, how much of the slide was analyzed. There was some mention of number of images but not exactly how many fields were covered. It's mentioned that one image is used for classification results.

3; In line 92, exclusion criteria are noted. Can additional explanation be given on the reasoning supporting the exclusion criteria?

Responses to the Reviewer

Reviewer #2 (Comments for the Author):

>The authors present a comprehensive study examining the performance of gram stains using a variety of artificial intelligence based methods. The study is well structured and written. A few questions:

Response: Thank you for your comments on our manuscript. We have responded to each one below.

1; What was the reasoning behind targeting gram stains from urine samples? Multiple studies have shown that routine gram stains from urine samples have limited clinical utility and in many settings, labs no longer routinely perform gram stains from urine samples.

Response: Thank you for your question. We used urine for gram staining because unlike sputum, the quality varies less, and the findings can be more readily linked to antibiotic selection. Furthermore, we considered it one of the most common infectious diseases, and the potential impact on appropriate antimicrobial use would be significant. As you noted, some reports indicate limited diagnostic utility of Gram staining of urine, while others describe its usefulness in reducing broad-spectrum antimicrobial use. We have added these points to the Introduction(line 74–80).

2; It is unclear from the document, how much of the slide was analyzed. There was some mention of number of images but not exactly how many fields were covered. Its mentioned that one image is used for classification results.

Response: The number of images corresponds to the number of fields of microscopic findings. In the manuscript, we have clarified that each image corresponds to one field of view (line 118). The number of images represents the number of available samples, collected according to the rules outlined in Table 2 and the rules outlined on lines 127–132.

3; In line 92, exclusion criteria are noted. Can additional explanation be given on the reasoning supporting the exclusion criteria?

Response: We apologize for the unclear reasoning. We have reviewed the exclusion criteria and realized that we made some mistakes. First, (1) and (2) are not separate; case (1) occurs when multiple microorganisms are identified in culture testing despite a single microorganism being

observed. Furthermore, case (2) occurs when the identified microorganism clearly differs from the microscopic finding. We have made the necessary corrections (lines 102–107). Specimens meeting criteria (1)–(3) were excluded to avoid misclassification, because the bacteria observed on Gram stain could not be reliably linked with a Class 2 organism (lines 107–110). The rationale has been added to the manuscript. Furthermore, since these specimens may be encountered in real-world practice, we have added their exclusion as a limitation of this study (lines 272–276).

Re: Spectrum03076-25R1 (Gram Staining Decipherment using an Artificial Intelligence-Powered Smartphone-Based Application)

Dear Dr. Kei Yamamoto:

Thank you for the privilege of reviewing your work. Below you will find my comments, instructions from the Spectrum editorial office, and the reviewer comments.

Please pay attention and address all the comments of reviewer #5 carefully as a final submission.

Revision Guidelines

Sincerely,
Aditya Padhi
Editor
Microbiology Spectrum

Reviewer #5 (Comments for the Author):

In their work titled "Gram Staining Decipherment using an Artificial Intelligence-Powered Smartphone-Based Application", the authors have developed computer aided diagnostic system to classify urine sample specimens. The manuscript has the scope of improvements. Following are my primary concerns/comments/queries.

Major Concern:

The authors are trying to develop a technology to diagnose urinary tract infection microbes. The authors used smartphones along with microscopic camera. So, taking pictures from smartphones can give different resolution of images based on distance of camera device and specimen. The authors have not discussed about these issues. Methods section does not mention about the algorithm used for training an Artificial intelligence-based model. This is one of the major concerns about the manuscript. Finally, it seems nothing new insight coming out of this manuscript.

Minor Concerns:

1. Which AI/ML algorithm was used to train the computer-aided diagnostic system (CAD).
2. Does the CAD system diagnoses the presence/absence of both, gram positive as well as gram negative bacteria in a particular sample?
3. Line No. 23: This line is misleading. Gram staining is not used for antimicrobial therapy. It is used for staining the bacterial colony leading to identification and classification of bacteria.
4. Line No. 27-28: How many species the CAD system classifies? What about presence of any new species? What morphology does the CAD system classify? e.g. round, cylindrical, spiral etc.
5. Line 34: The number of samples (433) is very less for any AI model training.
6. Line 72: It seems that the CAD system is an APP based on smartphone. Please clarify?
7. Line: 73: Which bacterial infection does it diagnose?
8. Line 87: Method section lacks discussion about the algorithm used in CAD system. This is one of the biggest drawbacks of this manuscript.
9. Line No 114-115: Please elaborate here, Does the model/system classify bacteria based on bacterial morphology? If so, what kind of morphologies can it classify? How was the AI system/model trained? How many types of morphologies were used for training the system/model?
10. Line 174 -170: This section can go into Methods section; these are the details about dataset. It's not result.
11. Line 178-179: Dataset 1 has 7940 images, while dataset 2 has only 2364 images, why? Both datasets were created from images obtained from 433 samples.

Reviewer #6 (Comments for the Author):

Based on the comments from the previous reviewer(s), the appropriate clarifications have been added to this manuscript. I have no further questions/concerns that need to be addressed.

In their work titled “**Gram Staining Decipherment using an Artificial Intelligence-Powered Smartphone-Based Application**”, the authors have developed computer aided diagnostic system to classify urine sample specimens. The manuscript has the scope of improvements.

Followings are my **primary** concerns/comments/queries.

Major Concern: The authors are trying to develop a technology to diagnose urinary tract infection microbes. The authors used smartphones along with microscopic camera. So, taking pictures from smartphones can give different resolution of images based on distance of camera device and specimen. The authors have not discussed about these issues. Methods section does not mention about the algorithm used for training an Artificial intelligence-based model. This is one of the major concerns about the manuscript. Finally, it seems nothing new insight coming out of this manuscript.

Minor Concerns:

1. Which AI/ML algorithm was used to train the computer-aided diagnostic system (CAD).
2. Does the CAD system diagnoses the presence/absence of both, gram positive as well as gram negative bacteria in a particular sample?
3. Line No. 23: This line is misleading. Gram staining is not used for antimicrobial therapy. It is used for staining the bacterial colony leading to identification and classification of bacteria.
4. Line No. 27-28: How many species the CAD system classifies? What about presence of any new species? What morphology does the CAD system classify? e.g. round, cylindrical, spiral etc.
5. Line 34: The number of samples (433) is very less for any AI model training.
6. Line 72: It seems that the CAD system is an APP based on smartphone. Please clarify?
7. Line: 73: Which bacterial infection does it diagnose?
8. Line 87: Method section lacks discussion about the algorithm used in CAD system. This is one of the biggest drawbacks of this manuscript.
9. Line No 114-115: Please elaborate here, Does the model/system classify bacteria based on bacterial morphology? If so, what kind of morphologies can it classify? How was the AI system/model trained? How many types of morphologies were used for training the system/model?
10. Line 174 -170: This section can go into Methods section; these are the details about dataset. It's not result.
11. Line 178-179: Dataset 1 has 7940 images, while dataset 2 has only 2364 images, why? Both datasets were created from images obtained from 433 samples.

Responses to the Reviewer

Reviewer #5 (Comments for the Author):

In their work titled "Gram Staining Decipherment using an Artificial Intelligence-Powered Smartphone-Based Application", the authors have developed computer aided diagnostic system to classify urine sample specimens. The manuscript has the scope of improvements. Followings are my primary concerns/comments/queries.

Major Concern:

We sincerely thank the reviewer for these important comments. We have addressed each concern below.

The authors are trying to develop a technology to diagnose urinary tract infection microbes. The authors used smartphones along with microscopic camera. So, taking pictures from smartphones can give different resolution of images based on distance of camera device and specimen. The authors have not discussed about these issues.

We agree that variability in camera positioning could potentially influence spatial resolution and image quality. To minimize such variability, all smartphone images were acquired under standardized magnification ($\times 1000$, oil immersion) using a NexyZ universal smartphone adapter attached directly to the microscope eyepiece. After optimal focus was achieved using the microscope, the visible microscopic field was maximized within the camera frame and images were captured using $1.9\times$ optical zoom. Although the exact camera-to-eyepiece distance was not quantitatively measured, the fixed adapter maintained a stable and reproducible camera position across acquisitions. This standardized acquisition protocol minimized variability related to camera positioning and ensured consistent field-of-view and comparable spatial resolutions across devices. This clarification has been moved from Appendices to the Methods section (Line 138-156). Furthermore, to address potential differences in spatial scale across imaging devices, a micrometer was used during image acquisition in Dataset 1 to standardize image size. However, this standardization did not result in a meaningful improvement in predictive performance. These findings suggest that, under standardized magnification, minor inter-device scale variations may have limited impact on classification accuracy.

Methods section does not mention about the algorithm used for training an Artificial

intelligence-based model. This is one of the major concerns about the manuscript.

We appreciate this important comment. In the Methods section of the revised manuscript, we have added a detailed description of the AI model architecture and training procedure (AI model architecture and training, Line 96-115).

Finally, it seems nothing new insight coming out of this manuscript.

We appreciate the reviewer's comment regarding the novelty of this study. However, to the best of our knowledge, this study represents one of the first evaluations of an AI system designed for smartphone deployment for the interpretation of Gram-stained urine specimens.

In this study, we:

- # Evaluated AI-based interpretation of Gram-stained urine images acquired using multiple imaging devices

- # Assessed the influence of different staining methods within the same AI framework

- # Examined the relative impact of staining methods and imaging devices on classification performance

- # Explored the influence of micrometer-based scale standardization on model accuracy

- # Performed confidence-stratified performance analysis using uncertainty estimation

Rather than focusing solely on algorithm development, this study aimed to evaluate practical factors affecting the interpretation of AI-assisted Gram staining results under real-world laboratory conditions. We have revised the Discussion section to more clearly highlight these implementation-oriented insights (Line 252-255).

Minor Concerns:

1. Which AI/ML algorithm was used to train the computer-aided diagnostic system (CAD).

We thank the reviewer for this important question. In the revised manuscript, we have moved the description of the AI/ML algorithm from the Appendix to the Methods section (Line 96-115). The CAD system is based on a deep learning convolutional neural network using the ConvNeXt architecture pretrained on ImageNet. We have clarified the training strategy, optimizer (AdamW), data augmentation (RandAugment and mixup), loss functions (multi-label evidential loss and evidential loss), cross-validation scheme, and Model Soups-based weight optimization.

2. Does the CAD system diagnoses the presence/absence of both, gram positive as well as gram negative bacteria in a particular sample?

Yes. The Class 1 model performs multi-label morphology classification and can detect the presence or absence of gram-positive cocci (GPC), gram-positive rods (GPR), gram-negative rods (GNR), gram-negative cocci (GNC), yeast, polymicrobial findings, or no bacteria (“None”). We have clarified this explicitly in the updated Methods section (Line 132-137).

3. Line No. 23: This line is misleading. Gram staining is not used for antimicrobial therapy. It is used for staining the bacterial colony leading to identification and classification of bacteria.

We thank the reviewer for this important comment. We agree that Gram staining is not used for antimicrobial therapy. The sentence has been revised to state that Gram staining provides rapid microbiological information that may assist in empirical antimicrobial selection.

4. Line No. 27-28: How many species the CAD system classifies? What about presence of any new species? What morphology does the CAD system classify? e.g. round, cylindrical, spiral etc.

We thank the reviewer for this helpful comment. We have revised both the Abstract (Lines 25–29) and Introduction to clearly describe the classification framework of the CAD system. The Class 1 model performs morphology-based classification into seven predefined categories (yeast, GPC, GPR, GNR, GNC, polymicrobial, and none). The Class 2 model classifies 17 predefined species-level categories, as detailed in Table 1. We have clarified that the system does not perform open-ended species identification but instead classifies samples according to predefined categories based on Gram-stain characteristics.

5. Line 34: The number of samples (433) is very less for any AI model training.

We respectfully clarify that 433 samples were used for validation in this study. The AI model was trained on 1,350 slides (13,901 images) collected outside the study period. This information has now been clearly described in the Methods section to avoid misunderstanding (Line 100-102).

6. Line 72: It seems that the CAD system is an APP based on smartphone. Please clarify?

We thank the reviewer for this important question. The CAD system is implemented as a smartphone

application that uses a cloud-based deep learning model. However, in the present study, inference was performed using the same cloud-based AI model in a PC environment to ensure standardized and device-independent evaluation. We have clarified this point in the revised manuscript (Line 97-99).

7. Line: 73: Which bacterial infection does it diagnose?

The CAD system does not directly diagnose a specific bacterial infection. However, it supports interpretation of urine Gram-stain findings in patients with suspected urinary tract infection by classifying bacterial morphology and selected species categories. We have clarified this in the Introduction sections (Line 72-74).

8. Line 87: Method section lacks discussion about the algorithm used in CAD system. This is one of the biggest drawbacks of this manuscript.

We agree with the reviewer and have substantially revised the Methods section accordingly. The AI architecture, training strategy, data splitting, augmentation, loss functions, cross-validation, and weight optimization procedures are now explicitly described in the main manuscript rather than only in the Appendix.

9. Line No 114-115: Please elaborate here, Does the model/system classify bacteria based on bacterial morphology? If so, what kind of morphologies can it classify? How was the AI system/model trained? How many types of morphologies were used for training the system/model?

Yes, the system classifies bacteria based on morphology in the Class 1 model. Seven morphology categories were used for training: yeast, GPC, GPR, GNR, GNC, no bacteria, and polymicrobial. The training dataset consisted of 1,350 slides (13,901 images), with slide-level data splitting to prevent leakage. These details have been added to the Methods section (Line 132-137).

10. Line 174 -170: This section can go into Methods section; these are the details about dataset. It's not result.

We thank the reviewer for this comment. We have moved the description of the eligible sample population (1,388 and 1,869 samples) to the Methods section to clarify the study design (Line 164-165). The final number of included samples remains in the Results section, consistent with

conventional reporting in clinical research.

11. Line 178-179: Dataset 1 has 7940 images, while dataset 2 has only 2364 images, why?

Both datasets were created from images obtained from 433 samples.

Dataset 1 included images captured using five imaging devices, whereas Dataset 2 included images captured using a single smartphone under two staining methods. Therefore, Dataset 1 contained more images due to multi-device acquisition. We have clarified this in the revised Methods section (Line 145-147).

Responses to the Reviewer

Reviewer #6

Based on the comments from the previous reviewer(s), the appropriate clarifications have been added to this manuscript. I have no further questions/concerns that need to be addressed.

We sincerely thank the reviewer for the careful evaluation of our revised manuscript and for the positive feedback. We appreciate the reviewer's time and effort.

Re: Spectrum03076-25R2 (Gram Staining Decipherment using an Artificial Intelligence-Powered Smartphone-Based Application)

Dear Dr. Kei Yamamoto:

Your manuscript has been accepted, and I am forwarding it to the ASM production staff for publication. Your paper will first be checked to make sure all elements meet the technical requirements. ASM staff will contact you if anything needs to be revised before copyediting and production can begin. Otherwise, you will be notified when your proofs are ready to be viewed.

Sincerely,
Aditya Padhi
Editor
Microbiology Spectrum

Reviewer #6 (Comments for the Author):

I believe that the authors have appropriately addressed the comments from the previous round of comments. I have no additional concerns.